# Mechanical Performance of Reclaimed Asphalt Pavement Modified with Waste Frying Oil and Crumb Rubber

**DOI:** 10.3390/ma14112781

**Published:** 2021-05-24

**Authors:** Munder Bilema, Mohamad Yusri Aman, Norhidayah Abdul Hassan, Zubair Ahmed Memon, Hend Ali Omar, Nur Izzi Md Yusoff, Abdalrhman Milad

**Affiliations:** 1Department of Highway and Traffic Engineering, Faculty of Civil Engineering and Built Environmental, Univesiti Tun Hussein Onn Malaysia, Parit Raja 86400, Batu Pahat, Johor, Malaysia; mdyusri@uthm.edu.my; 2Department of Geotechnics & Transportation, School of Civil Engineering, Faculty of Engineering, Universiti Teknologi Malaysia, Johor Bahru 81310, Johor, Malaysia; hnorhidayah@utm.my; 3Department of Engineering Management, College of Engineering, Prince Sultan University (PSU), Riyadh 11586, Saudi Arabia; zamemon@psu.edu.sa; 4Department of Civil Engineering, University of Tripoli, 22131 Tripoli, Libya; hend.omar@uot.edu.ly; 5Department of Civil Engineering, Universiti Kebangsaan Malaysia, UKM Bangi 43600, Selangor, Malaysia; izzi@ukm.edu.my

**Keywords:** recycling, RAP, mechanical performance, rejuvenator, waste frying oil, crumb rubber, rutting resistance, moisture damage

## Abstract

Researchers are exploring the utilisation of reclaimed asphalt pavement (RAP) as a recycled material to determine the performance of non-renewable natural aggregates and other road products such as asphalt binder, in the construction and rehabilitation stage of asphalt pavements. The addition of RAP in asphalt mixtures is a complex process and there is a need to understand the design of the asphalt mixture. Some of the problems associated with adding RAP to asphalt mixtures are moisture damage and cracking damage caused by poor adhesion between the aggregates and asphalt binder. There is a need to add rejuvenators to the recycled mixture containing RAP to enhance its performance, excepting the rutting resistance. This study sought to improve asphalt mixture performance and mechanism by adding waste frying oil (WFO) and crumb rubber (CR) to 25 and 40% of the RAP content. Moreover, the utilisation of CR and WFO improved pavement sustainability and rutting performance. In addition, this study prepared five asphalt mixture samples and compared their stiffness, moisture damage and rutting resistance with the virgin asphalt. The results showed enhanced stiffness and rutting resistance of the RAP but lower moisture resistance. The addition of WFO and CR restored the RAP properties and produced rutting resistance, moisture damage and stiffness, which were comparable to the virgin asphalt mixture. All waste and virgin materials produce homogeneous asphalt mixtures, which influence the asphalt mixture performance. The addition of a high amount of WFO and a small amount of CR enhanced pavement sustainability and rutting performance.

## 1. Introduction

The asphalt industry is using increasing amounts of reclaimed asphalt pavement (RAP) to reduce the vastly abundant reusable asphalt pavement recycled materials and adopt environment-friendly production processes, without compromising asphalt pavement performance [1]. However, RAP has a higher amount of fine aggregates than the old hot-mix asphalt (HMA), due to the milling process. RAP binders aged through oxidation and loss of volatiles, resulting in a brittle and stiff binder. The presence of aged binder in RAP raises the concern for the vulnerability of the RAP in asphalt mixtures to thermal and fatigue cracking [2,3].

Previous research on the effects of RAP on asphalt mixtures found that RAP could improve rutting resistance and stiffness, while reducing the moisture and fatigue resistance of the asphalt mixtures [4,5,6]. However, the high RAP fraction in the asphalt mixtures may affect pavement performance [7,8]. One serious concern is the lack of adhesion between the aggregates and asphalt binder, which affects aggregate binding and could lead to cracking and substantial pavement disintegration. Binder stiffness could pose a problem in field compaction and lead to premature field failure and pavement cracking [9]. Oxidation may occur due to the chemical composition changes during the mixing process, and the ratio of maltenes to asphaltenes in the asphalt binder is a concern when RAP is applied with a proper rejuvenation dosage material [10].

Temperature is a critical factor in a chemical reaction that causes severe asphalt oxidation, where extensive oxidation during mixing causes binder hardening and loss of volatile compounds [11,12]. Oxidation also occurs during the production and transportation of asphalt mixture to the paving site, at slow placement and compaction rates. Once the asphalt pavement is opened to traffic, the age-hardening process continues at a slow rate, depending on the pavement condition and climatic factors like temperature and moisture [13]. The oily components in the asphalt structure system are lost through volatility or absorption by aggregates. This results in excessive hardening and embrittlement, which eventually causes a decline in the binder performance [14,15].

Therefore, it is essential to restore the original asphaltene to maltene ratio. Adding waste frying oil (WFO) to HMA improves the workability of the asphalt binder [16]. Adding RAP and WFO to HMA contributes to the alternatives available for replacing natural resources [17]. Using WFO as a rejuvenator reduces the oxidation level and restores the oily components of the aged binder [18,19]. Using WFO as a rejuvenator produces a softer RAP binder and reduces the rutting resistance of asphalt [20,21,22]. The addition of rejuvenators could increase the maltene-to-asphaltene ratio and reduce viscosity and yield, which causes permanent deformation problems [23,24]. Based on the studies on the effect of rejuvenators on asphalt mixture, Kaseer et al. [25] concluded that adding waste vegetable oil in asphalt mixtures reduced the rutting resistance and the resilient modulus through the rise in waste vegetable oil dosage in the asphalt mixture. Garcia et al. [26] found that sunflower oil reduced the resilient modulus and the indirect tensile strength (ITS) of asphalt mixtures. Some researchers concluded that adding rejuvenators to asphalt mixtures could reduce the ITS and rutting resistance [27,28,29]. It is essential to reduce severe oxidation by improving the viscosity of structural asphalt binder. One way to enhance binder viscosity is by incorporating crumb rubber (CR) into the asphalt mixture, to harden the asphalt and produce the required workability at high temperatures. Previous studies reported that adding CR to the asphalt mixture enhanced the durability and stiffness of the asphalt, while increasing its lifespan [30,31].

Previous studies have determined the optimum content of the recycling agent required to balance the performance characteristics of high-RAP mixtures, except for the rutting resistance. Therefore, this study used crumb rubber to increase the rutting resistance caused by the addition of a rejuvenator and found that 1.5% CR improved the overall performance of the asphalt mixture [32,33]. Bilema et al. [34] investigated the considerable effect of CR on the ITS of the asphalt mixture stiffness. Using discarded vehicle tires in pavement construction has environmental benefits and prevents tire stockpile or disposal in the landfill [35,36]. There are two reasons for adding RAP and WFO, together with CR, in HMA, namely to minimise the utilisation of natural resources and discover a sustainable pavement alternative. Therefore, the primary aim of this study was to assess the effect of WFO and CR on RAP behaviour.

## 2. Experimental Design

### 2.1. Materials

The RAP was collected from the E2 North-South Expressway, Johor, Malaysia, between 140 and 148 km. The Kemaman Bitumen Company Sdn. Bhd (Selangor, Malaysia) supplied the virgin asphalt binder (grade 60/70). The WFO used to fry French fries was collected from a local restaurant and pre-treated using a simple filtering process, with a filtration paper of 150-mm diameter. Miroad Rubber Industry, Johor, Malaysia supplied the CR with a size of 0.15 mm (mesh 40). Table 1 shows the design matrix for the asphalt mixture.

The asphalt mixture was prepared following the Superpave asphalt mixture design system. The design of all asphalt mixtures followed the NMAS 12.5 mm, to determine the optimum asphalt binder content (OBC). The aggregate gradation was between the upper and lower limits to fulfil the requirement for Superpave gradation. The asphalt binder content of the HMA followed the AASHTO T-308 and was determined using the ignition method [37]. The separation of the asphalt binder from the aggregate was achieved at a high temperature of 538 ± 5 °C. Two samples were tested for fine and coarse aggregates.

#### 2.1.1. WFO and CR Content

This study performed the penetration, softening point, ductility, and viscosity tests to determine the WFO and CR contents. Table 2, Table 3, Table 4 and Table 5 show the results of the trials for obtaining the WFO and CR percentages. The percentages of the WFO were chosen on the basis of the physical tests with a lower viscosity result. On the other hand, the percentages of the CR were chosen, depending on the stiffness with appropriate workability.

#### 2.1.2. Virgin Aggregate Tests

Table 6 shows that 11.3% of the aggregate impact value fulfilled the requirement for the mix design standard. A total of 20% of Los Angeles abrasion fulfilled the required range for the Superpave mix design of 35% to 45%. The study performed the fine aggregate angularity, sand equivalent and harmful material tests to prevent the rutting, shoving and stripping of the asphalt mixtures. These tests ensured that the aggregate did not contain spherical particles, clay and wood shale, all of which could affect the asphalt mixture performance. The 53.5%, 56% and 0.27% results fulfilled the specifications for fine aggregate angularity, sand equivalent and harmful materials. The flat and elongated particles in the coarse aggregate test were determined to ensure they did not affect the compaction process.

The 3.89% flat and elongated particles were below the 10% maximum limit for the Superpave standard. Therefore, the virgin aggregates proposed by this study were suitable for the Superpave mix design.

AASHTO T85 performed the aggregate’s specific gravity for the coarse aggregate size, while AASHTO T84 performed this process for the fine size aggregate [46]. The bulk-specific gravity is a vital aggregate characteristic in computing the volumetric properties of the mixtures, such as the percentage of voids in mineral aggregate (VMA) and air voids (Va). The 2.68 specific gravity of the virgin aggregates used in this study was within the 2.40 to 3.00 recommended range for the Superpave mix design.

#### 2.1.3. Aggregate Gradation

The aggregate gradation followed the Superpave grading system with a 12.5 NMAS and 0.45 power grading chart for the dense-graded mix design. Furthermore, the sieve sizes used in this study were as follows—19, 12.5, 9.5, 4.75, 2.36, 1.18, 0.600, 0.300, 0.15 and 0.075 mm and the pan. Table 7 presents the gradation for the 1200 and 2200 g asphalt mixture samples. The aggregates were heated in a 60 °C oven for 24 h before sieving, to remove the moisture.

#### 2.1.4. Preheating the RAP

The milled RAP was collected from the construction site. The RAP has a high moisture content that led to low HMA quality. A total of 500 g aggregates was heated in a 105 °C oven. The sample was tested every 30 min to obtain minimum moisture content with minimum heating time and prevent moisture damage in the asphalt mixture. Overheating the RAP could produce brittle aggregate, which results in reduced stiffness and durability of the asphalt mixture [9].

Figure 1 shows the change in the moisture content of the RAP, where longer heating of the RAP resulted in lower moisture content. Heating the RAP for one hour produced 0.32% moisture content and did not affect the asphalt mixture. The RAP exposed to pressure and high temperature during its service life has lower durability than the virgin aggregate. A shorter heating period of the RAP is more suitable, since overheating increases the RAP stiffness. This study adopted a heating period of one hour for all RAP content.

#### 2.1.5. Binder Content of the RAP

Table 8 presents the asphalt binder content of the RAP. The fine aggregates contained more asphalt binders than the coarse aggregates because their high surface area absorbed more asphalt binders. The result for the asphalt binder was similar to the actual asphalt binder content of the construction company, with a 0.02% difference between the calculated and actual asphalt binder contents.

#### 2.1.6. Optimum Asphalt Binder Content

The optimum asphalt binder content (OBC) of the specimen was established using the 12.5 nominal maximum aggregate size (NMAS). The OBC for five asphalt mixtures (Virgin 60/70, 25% RAP, 25% RAP + WFO + CR, 40% RAP and 40% RAP + WFO + CR) was determined following the AASHTO T312 procedures published by the Asphalt Institute for the Superpave mix design [47]. The design mixture was based on the medium-to-high traffic-load category, equivalent to three or lower than 30 million ESALs. The primary target of this mixture was to achieve 4% Va for all samples. The trial blends were selected as the design aggregate structure to determine the OBC for this study. The specimens were mixed and compacted at varying asphalt binder contents. The five selected trials OBCs were 4.5, 5.0, 5.5, 6.0 and 6.5%. The properties of the trial blends were determined and tabulated, where the parameters for the asphalt mixture were air voids (Va), voids in the mineral aggregate (VMA), and voids filled with asphalt (VFA) [48]. Table 9 presents the results for the OBC.

### 2.2. Asphalt Mixture Performance

The study determined the asphalt mixture performance to predict the asphalt mixture’s behaviour and compare the performance of the HMA pavements containing the recycled mixture and virgin mixture, and their mechanical properties. A total of 105 specimens for tests were used. These included 3 specimens for the ITS test, 6 specimens for the moisture sensitivity tests (3 for conditions specimens (Wet), 3 for the uncondition (Dry) specimens, 3 specimens for dynamic creep and 6 specimens for the resilient modulus test (3 specimens for 25 °C and 3 specimens for 40 °C). A total of 110 specimens were used for the process of OBC, air voids and test conditions.

#### 2.2.1. Indirect Tensile Strength (ITS)

The cracking potential of a mixture was determined on the basis of the tensile failure strain. This test measured the asphalt mixture resistance to tensile strength. The test was conducted at room temperature (25 °C). The failure load was then recorded, and the ITS was calculated using the following equation.
(1)$$\mathrm{ITS}\;=\frac{2000F}{\pi \mathrm{tD}\;}\;$$
where ITS is the indirect tensile strength (kPa), F is failure load (N), t is the sample height (mm) and D is the sample diameter (mm).

#### 2.2.2. Moisture Susceptibility Test

The moisture sensitivity test quantifies the HMA mixture’s ability to resist water damage regardless of the source, and determines the degree of moisture damage. The test followed the AASHTO T283 (2007): Resistance of Compacted Bituminous Mixture to Moisture-Induced Damage [49]. Two sets of compacted samples (100 mm diameter and 75 mm long) were subjected to a split tensile test or tensile strength ratio test, where one sample set served as the control. The other sample set was placed in a partial vacuum and soaked in water for 24 h. The tensile strength ratio (TSR), which is the ratio of the average split tensile strength of the conditioned (wet) sample over the average split tensile strength of the unconditioned (dry) sample, must comply with the Superpave^®^ requirement (TSR ≥ 0.80) to prevent the possible moisture-induced problems [35]. Table 10, shows the moisture sensitivity test factors.

Five asphalt mixture samples (100 mm diameter and 75 mm length) were prepared. After mixing, the samples were allowed to cool at room temperature for two hours. The samples were cured in the oven at 60 °C for 16 h and 135 °C for two hours, before compaction using the Superpave Gyratory Compactor (SGC, Servopac Gyratory Compactor, Victoria, Australia). The compacted samples were stored at room temperature (25 °C) for 72 to 96 h. Following this, the samples were divided into two sets—unconditioned (dry) samples and conditioned (wet) samples. The unconditioned samples were wrapped in plastic and stored at room temperature.

The indirect tensile strength ratio values were calculated, using the following equation.
(2)$$S_{t}=\frac{2000P}{\pi \mathrm{tD}\;}\;$$
where S_t_ is the tensile strength (kPa), P is the maximum load (N), t is sample thickness (mm) and D is sample diameter (mm). The resistance to moisture damage is a ratio of the unconditioned sample tensile strength retained after the conditioning. The TSR was calculated using the following equation.
(3)$$\mathrm{TSR}\;=\frac{S_{2}}{S_{1}}\;$$
where TSR is the tensile strength ratio (>80%), S_1_ is the average tensile strength of the unconditioned samples, and S_2_ is the average tensile strength of the conditioned samples.

#### 2.2.3. Resilient Modulus Test

This study performed the resilient modulus tests (ASTM D4123) at 25 °C (resistance to cracking) and 40 °C (resistance to rutting) [50]. Table 11 presents the parameters of resilient modulus. In most cases, the resilient modulus of asphalt mixtures dropped significantly at a higher temperature. The strain was measured using a linear vertical differential transducer (LVDT).

#### 2.2.4. Dynamic Creep Test

This study performed the dynamic creep test (NCHRP 9-19) to determine the permanent deformation of the virgin asphalt and the recycled asphalt mixtures. Table 12 presents the parameters for the dynamic creep test.

#### 2.2.5. Wheel Tracking Test

This study used the Wessex wheel tracking device in the wheel tracking test, to assess the relation between resistance and rutting and the passing wheel simulation for the pavement. Table 13 presents the parameters of the wheel tracking test. The test involved 2200 g of the compacted mixture (air void content of 5 ± 0.5%), following the testing criteria for a dimension of 300 mm diameter and height of 65 ± 1 mm. The samples were conditioned at the testing temperature for at least six hours, and the wheel tracker was preheated for one hour at the testing temperature. The testing temperature was within the acceptable standard temperature range before commencing the test.

## 3. Results and Discussion

### 3.1. Performance Tests

#### 3.1.1. ITS

The specimens for the ITS test were prepared with 4% Va, and the test was performed following the AASHTO T283. Figure 2 presents the ITS values at 25 °C, where the ITS for the asphalt mixture containing 25% and 40% RAP was 27.8% and 42.8% higher than the virgin asphalt mixture. The higher ITS values indicate the higher stiffness and higher viscosity of the RAP asphalt binders. This result is similar to those obtained by Idham et al. [51], who concluded that the RAP has a significant impact on the ITS, where higher RAP contents produced higher ITS values.

The ITS value of the CR and WFO-modified asphalt mixtures containing 25% RAP and 40% RAP was 713 and 686 kPa. The lower ITS values were due to the addition of CR and WFO in the RAP asphalt mixture. The asphalt mixtures containing 25% RAP, CR and WFO had an ITS of 31.1% as compared to the asphalt mixture with 25% RAP. The ITS value of the asphalt mixture containing 40% RAP, CR and WFO was 47.3% lower than the asphalt mixture with 40% RAP. WFO reduced the ITS value through the lower viscosity of the aged asphalt binder in the RAP. The ITS value of the asphalt mixture containing CR, WFO and RAP was similar to the virgin asphalt. This result was consistent with Eriskin et al. [17], which showed that the WFO reduced the ITS value.

The ITS results were analysed using paired sample *t*-test to compare the significance of the group means of the virgin asphalt containing RAP and the recycled asphalt mixtures. Table 14 shows a significant difference in the ITS results. There was a statistically significant difference between the virgin asphalt and asphalt mixtures with 25% and 40% RAP contents (*p* < 0.05), which indicates the effect of adding RAP on the strength of the asphalt mixture. However, there was no statistically significant difference between the virgin asphalt and asphalt mixtures containing WFO, CR and 25% RAP and 40% RAP (*p* > 0.05), which indicates that WFO and CR reduced the hardness of the RAP asphalt mixture. In summary, the ITS for the asphalt mixtures containing WFO, CR and 25% RAP and 40% RAP was comparable to the ITS of the virgin asphalt.

#### 3.1.2. Moisture Sensitivity

The tensile strength ratio (TSR) is the ITS of the wet value divided by the dry value; it measures the moisture damage in the asphalt mixture. Moisture sensitivity is generally based on the cohesive resistance between the aggregates and the asphalt binder [35]. Figure 3 shows the TSR for the virgin asphalt, RAP and recycled asphalt mixtures. The lower ITS values in the moisture sensitivity test were compared with the ITS related to high Va 7 ± 0.5%. The ITS of the wet or conditioned samples was lower than the dry or unconditioned samples, due to water presence and high Va. Figure 3 shows that the TSR for the asphalt mixture with 25% RAP and 40% RAP was lower than the virgin asphalt mixture. The aged asphalt binder also lost its ability to interlock the aggregate with asphalt binder, due to its high viscosity during its lifespan. The low TSR of 4.6% and 10.2% of the 25% RAP and 40% RAP was comparable to the virgin asphalt.

The asphalt mixture with 25% RAP had a TSR of 80.1%, which fulfilled the specification limit of 80%. The asphalt mixture with 40% RAP had a TSR of 74.5% and did not meet the criteria limit of the moisture sensitivity test. Adding RAP to the asphalt mixture increased the TSR values, which was consistent with the findings by Peralta et al. [52]. Adding WFO and CR enhanced the moisture resistance and increased the TSR of the recycled asphalt mixture. According to Wen et al. [53], the addition of WFO into the RAP asphalt mixture significantly increased the TSR result, contributing to similar results to the virgin asphalt mixture concerning the dosage of the WFO. The asphalt mixture containing WFO, CR and 25% RAP showed a 5.1% increase in TSR relative to the asphalt mixture with 25% RAP. The asphalt mixture containing WFO, CR and 40% RAP showed a 7% increase in TSR, relative to the asphalt mixture with 40% RAP. The higher TSR could be due to the combined effect of WFO and CR in enhancing the cohesion between the aggregate and asphalt binder and the adhesion between the aged and virgin asphalt binder, which resulted in better moisture resistance of the recycled asphalt mixture. The asphalt mixture with 25% RAP and 40% RAP and the combination of CR and WFO led to an increase in the TSR, which caused similar results to virgin asphalt mixture with 85.2% and 81.5%, respectively.

The fatty acid in the WFO was one of the factors contributing to the higher moisture resistance. Overall, these results are consistent with those of a recent study by Foroutan et al. [24]. The WFO and CR have a combined effect of increasing the TSR, which means higher moisture resistance. Adding WFO increased the coating ability of the RAP binder and thus increased the moisture resistance of the asphalt mixture. Therefore, the combined use of WFO and CR with RAP could restore the asphalt mixture performance.

This study performed the paired *t*-test to compare the significant difference between the virgin asphalt with RAP and recycled asphalt mixtures. Table 15 presents the differences between asphalt mixtures in terms of moisture sensitivity. There were statistically significant differences between the virgin asphalt and 40% RAP asphalt mixture (*p* = 0.038 and *p* > 0.05). Even though adding 40% RAP to the asphalt mixture increased the moisture damage, there was no significant change in the virgin asphalt with 25% RAP, virgin asphalt with WFO, CR and 25% RAP, and virgin asphalt with WFO, CR and 40% RAP with *p* = 0.269, *p* = 0.341 and *p* = 0.499, respectively. In summary, 25% RAP content, 25% RAP content with WFO and CR and 40% RAP content with WFO and CR, presented no significant differences as compared to the virgin asphalt mixture.

#### 3.1.3. Resilient Modulus

The resilient modulus is a critical factor in asphalt mixture stiffness, as it indicates the quality of the asphalt mixture. Figure 4 presents the resilient modulus for the virgin asphalt, RAP and the recycled asphalt mixtures at 25 and 40 °C. The RAP in the asphalt mixture has higher resilient modulus values than the virgin asphalt at both temperatures. The values for the resilient modulus of all asphalt mixtures decreased as the test temperature increased from 25 to 40 °C. According to Poulikakos et al. [15], higher test temperatures resulted in lower stiffness of the asphalt mixture.

The higher resilient modulus at 25 °C indicates that the asphalt mixture containing RAP had higher stiffness than the virgin asphalt. The asphalt mixtures containing 25% and 40% RAP had a resilient modulus of 8390 and 9584 MPa, which was 44.4% and 51% higher than the virgin asphalt. Izaks et al. [6] obtained similar results, where adding higher RAP percentages in the asphalt mixture resulted in higher stiffness and resilient modulus.

The CR and WFO have a combined effect of reducing the resilient modulus at 25 °C. Figure 4 shows that the asphalt mixture containing CR, WFO and 25% RAP has a resilient modulus of 4571 MPa, which was 45.5% lower than the asphalt mixture with 25% RAP. The asphalt mixture containing CR, WFO and 40% RAP had a resilient modulus of 4732 MPa, which was 50.6% lower than the asphalt mixture with 40% RAP.

The addition of oil usually reduced the asphalt mixture stiffness [26]. The reduced stiffness indicated the combined effect of WFO and CR on the asphalt mixture stiffness. The asphalt mixtures containing CR, WFO and 25% and 40% RAP had a resilient modulus value comparable to the virgin asphalt at 25 °C. The lower stiffness of the RAP asphalt mixture was due to the effect of WFO softening the RAP asphalt binder.

Table 16 presents the significant differences in the group means of the resilient modulus of the virgin asphalt, RAP and the recycled asphalt mixtures at 25 °C. The table shows the statistically significant difference between the virgin asphalt and asphalt mixture with 25% RAP and 40% RAP (*p* < 0.05). Adding the RAP increased the stiffness of the asphalt mixture relative to the virgin asphalt. However, there was no significant difference between the virgin asphalt with 25% RAP and 40% RAP, and the rejuvenated rubber and CR with *p* = 0.357 and *p* = 0.315, respectively.

The results showed that the resilient modulus increased with a higher RAP content at 40 °C, relative to the virgin asphalt. The replacement of virgin asphalt with 25% RAP and 40% RAP resulted in a 47.7% and 63% higher resilient modulus. The asphalt mixtures with 25% RAP and 40% RAP had the highest resilient modulus of 1701 and 2413 MPa at 40 °C. In contrast, WFO and CR reduced the resilient modulus values at 40 °C. The resilient modulus of the asphalt mixtures with CR, WFO and 25% RAP and 40% RAP was 848 and 901 MPa, which was similar to the 889 MPa resilient modulus of the virgin asphalt at 40 °C. The asphalt mixture with WFO, CR and 25% RAP had a 50% lower resilient modulus than the asphalt mixture with 25% RAP. The resilient modulus of the asphalt mixture with WFO, CR and 40% RAP was 62.6% lower than the asphalt mixture with 40% RAP. The combined use of CR and WFO could restore the RAP asphalt mixture performance. Therefore, adding oil to the RAP asphalt mixture could reduce the resilient modulus value at different temperatures [24].

The resilient modulus of the mixtures at 40 °C was compared using the paired *t*-test statistical analysis to determine the significant difference between the virgin asphalt with RAP and the recycled asphalt mixtures. Table 17 shows the significant difference in the resilient modulus at 40 °C. There was a marked difference between the virgin asphalt and the asphalt mixture with 25% RAP and 40% RAP (*p* < 0.05), which indicates that the addition of RAP increased the stiffness of the asphalt mixtures. However, there is no difference in the resilient modulus of the virgin asphalt and the recycled asphalt mixture (*p* > 0.05). Therefore, WFO and CR reduced the stiffness of the RAP asphalt mixture and retained a stiffness similar to the virgin asphalt.

#### 3.1.4. Dynamic Creep

This study performed the dynamic creep test to determine the permanent deformation and rutting of the asphalt mixture at 40 °C. Figure 5 shows the permanent deformation for the virgin, RAP and the recycled asphalt mixtures. Adding RAP to the asphalt mixture reduced the permanent deformation and improved the rutting resistance. Previous research has shown that RAP could increase the rutting resistance [28]. The lower permanent deformation of the RAP asphalt mixtures was associated with the high stiffness of the aged asphalt binder, while the high viscosity improved its rutting resistance. The asphalt mixture with 25% RAP showed reduced permanent deformation of 62.8%, relative to the virgin asphalt.

The asphalt mixture with 40% RAP showed a 79.2% reduction in permanent deformation relative to the virgin asphalt. In contrast, the permanent deformation of the asphalt mixtures with 25% RAP and 40% RAP was 0.193 and 0.108 mm at 40 °C. These results are consistent with those obtained by Pasetto et al. [54], who concluded that the RAP efficiently reduced the excessive permanent deformation.

The asphalt mixtures with CR, WFO and 25% RAP and 40% RAP showed higher permanent deformation of 60.5% and 77%, relative to the asphalt mixture with 25% RAP and 40% RAP. The asphalt mixtures with CR, WFO and 25% RAP and 40% RAP had a permanent deformation of 0.489 and 0.471 mm at 40 °C. Figure 5 shows that WFO could reduce the hardness of the RAP asphalt mixture, which increased its permanent deformation. Therefore, WFO and CR increased the permanent deformation of the RAP and produced an asphalt mixture with similar properties as the virgin asphalt.

Table 18 summarises the analysis using the paired *t*-test that compared the virgin asphalt with RAP and recycled asphalt mixtures. All asphalt mixtures containing RAP were significant, relative to the virgin asphalt (*p* < 0.05). There was no statistically significant difference between the virgin asphalt and the recycled asphalt mixtures with WFO and CR (*p* > 0.05). The results of this analysis demonstrate the ability of WFO and CR to recycle the RAP.

#### 3.1.5. Wheel Tracking

Wheel tracking gives an accurate prediction of the rut depth of the asphalt mixture. One feature of the truck wheel is its ability to predict the rutting depth at various temperatures, which allows for a better field simulation. In countries with a warm climate, there is a need to assess asphalt mixtures for permanent deformation resistance at high temperatures. Figure 6 shows the rutting depth of all asphalt binders at various temperatures.

Figure 6 shows that adding 25% RAP to the asphalt mixture could reduce the asphalt mixture’s rut depth at 45 °C. The asphalt mixture with 40% RAP had the lowest rut depth relative to all other asphalt mixtures at 45 °C, due to the large quantities of the carbonyl group in the RAP asphalt binder. This behaviour is consistent with the findings by Xinxin et al. [55], which showed that the high quantities of carbonyl in the aged asphalt binder reduced the rutting depth. The asphalt mixtures with 25% RAP and 40% RAP had a rut depth of 2.58 and 2.02 mm. The rut depth decreased with a higher percentage of the RAP content in the asphalt mixture. In contrast, the asphalt mixtures with CR, WFO and 25% RAP and 40% RAP had a comparable rut depth as the 60/70 virgin asphalt.

The asphalt mixtures with CR, WFO and 25% RAP and 40% RAP had a rut depth of 3.36 and 3.33 mm. Adding WFO reduced the viscosity of the RAP asphalt mixture, which indicates reduced rutting resistance. This result is consistent with the research by Zaumanis et al. [20], where the addition of WFO increased the rutting potential of the RAP asphalt mixture. The increase in the rut depth with asphalt mixtures with CR, WFO and 25% RAP and 40% RAP was due to the WFO, which softened the RAP asphalt mixture.

The rutting at 45 °C was analysed using the paired sample *t*-test, to compare the significance of group means of the virgin asphalt with the RAP and recycled asphalt mixtures. Table 19 shows the significant differences in wheel tracking results at 45 °C. There was a statistically significant difference between the virgin asphalt and the asphalt mixture with 25% RAP and 40% RAP with *p* = 0.037 and *p* = 0.002. Adding 25% RAP had a minor impact on the rutting depth, as the analysis result showed that the *p*-value equalled 0.037. On the contrary, there was no statistically significant difference between the virgin asphalt and asphalt mixture with CR, WFO and 25% RAP and 40% RAP (*p* > 0.05), which indicates that WFO and CR softened the RAP asphalt mixture.

Figure 6 shows that, at a test temperature of 60 °C, there was a higher possibility of asphalt mixture rutting. The asphalt mixtures with 25% RAP and 40% RAP had a rut depth of 3.64 and 2.92 mm, which was lower than the 60/70 virgin asphalt. Adding CR, WFO and 25% RAP or 40% RAP increased the rut depth of the asphalt mixtures. The greater rut depth was due to the addition of oil, which was consistent with the findings of previous studies [21,24] that concluded that oil increased the maltene-to-asphaltene ratio, reduced hardness and softened the aged asphalt mixture. The asphalt mixtures with CR, WFO and 25% RAP and 40% RAP had a rut depth of 4.49 and 4.45 mm. Therefore, all asphalt mixtures in this study fulfilled the rutting failure criteria with values lower than 15 mm. Figure 3 and Figure 6 show the inverse relationship between the wheel tracking and moisture damage results when using WFO as a rejuvenator. This result was consistent with a previous study by Yang et al. [56] that discovered that a higher moisture sensitivity produced lower rutting resistance. Figure 5 and Figure 6 show a direct correlation between wheel tracking and permanent deformation, where all asphalt mixtures with WFO and CR showed the same pattern of improved rutting resistance.

Table 20 shows the significant difference in the group means of the wheel tracking of the virgin asphalt with RAP and the recycled asphalt mixture at 60 °C. Table 20 shows the statistically significant differences between the virgin asphalt and the asphalt mixture with 25% RAP and 40% RAP, with *p* = 0.017 and *p* = 0.006. The analysis showed that RAP improved the rutting resistance. However, there was no significant difference between the virgin asphalt and asphalt mixture with WFO, CR and 25% RAP and 40% RAP with *p* = 0.713 and *p* = 0.443.

## 4. Conclusions

Based on the results of this study, the following conclusions are drawn.

The ITS increased with a higher RAP content in the asphalt mixture. Recycled asphalt mixtures that contain waste materials had an ITS similar to the virgin asphalt mixture, due to the effect of the WFO.The asphalt mixtures with 25% RAP and 40% RAP reduced the TSR, while the asphalt with 25% RAP fulfilled the TSR specification. The recycled asphalt mixtures containing waste materials had a higher tensile strength ratio, indicating a higher moisture resistance.The resilient modulus for asphalt with 25% RAP and 40% RAP was higher than virgin asphalt. The pattern for resilient modulus for all recycled asphalt containing both WFO and CR was identical at 25 and 40 °C. The asphalt mixtures with 25% RAP and 40% RAP had the highest resilient modulus.Asphalt mixtures with 25% RAP and 40% RAP showed an enhanced deformation performance relative to the virgin asphalt. The asphalt mixture containing the waste materials also had a lower permanent deformation than the virgin asphalt.The rutting values of all asphalt mixtures in the wheel tracking test had the same pattern at 45 and 60 °C test temperatures. The rutting resistance of the recycled asphalt containing waste materials was slightly improved, as compared to virgin asphalt at 60 °C.The CR and WFO complemented one another. The WFO restored the recycled asphalt pavement properties, and the CR improved the rutting resistance of the asphalt mixtures and gave them better workability. Therefore, the combined use of WFO and CR in recycled asphalt mixtures could improve moisture resistance, stiffness and rutting resistance.

## Figures and Tables

**Figure 1 materials-14-02781-f001:**
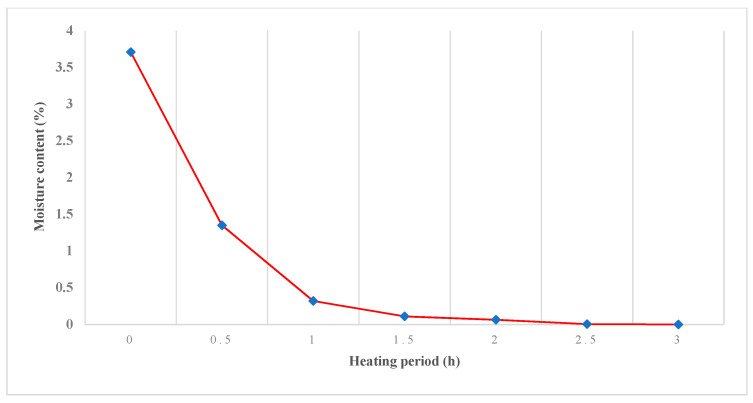
The effect of the heating time on the RAP moisture content.

**Figure 2 materials-14-02781-f002:**
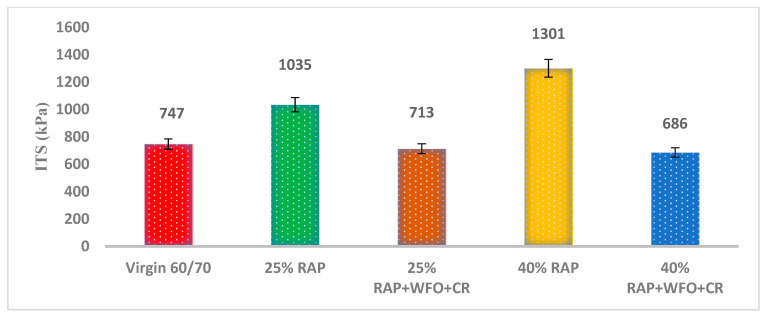
ITS for the virgin, RAP and recycled asphalt mixture at 25 °C.

**Figure 3 materials-14-02781-f003:**
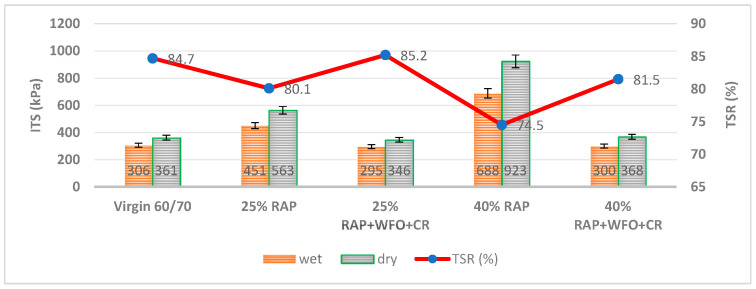
TSR for the virgin, RAP, and recycled asphalt mixtures.

**Figure 4 materials-14-02781-f004:**
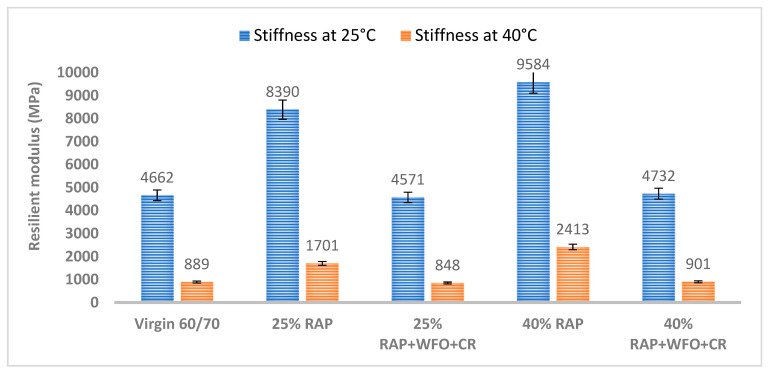
Resilient modulus of the virgin asphalt, RAP and the recycled asphalt mixture at different temperatures.

**Figure 5 materials-14-02781-f005:**
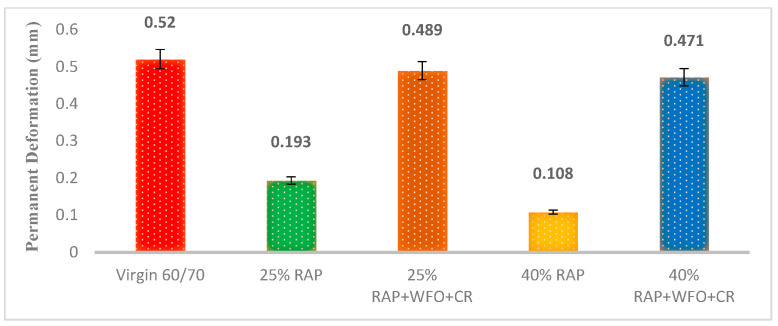
The permanent deformation of all samples at 40 °C.

**Figure 6 materials-14-02781-f006:**
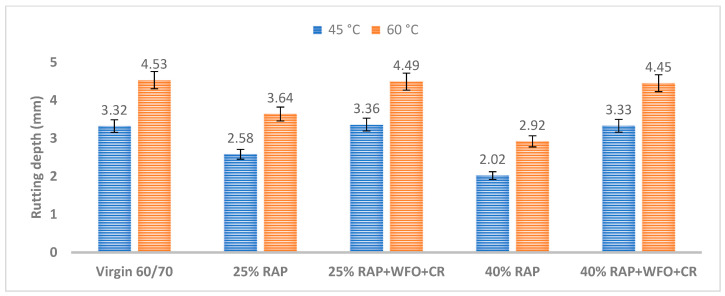
The rutting of all samples at 45 and 60 °C.

**Table 1 materials-14-02781-t001:** Design matrix of the asphalt mixtures.

No.	Factor	Details
1	Mix design method	Superpave mix design method
2	Aggregate gradation	All gradation satisfy the Superpave graduation requirements with 12.5 NMAS
3	Type of asphalt binder	Penetration grade (PEN) 60–70
4	WFO content	2.7 and 4.6%
5	RAP content	25 and 40%
6	CR content	1.5%

**Table 2 materials-14-02781-t002:** Trials to determine the optimum waste frying oil content for the 25% RAPB.

WFO. (%)	Penetration (PEN)	Softening Point (°C)	Ductility (cm)	Viscosity at 135 °C and 165 °C (cp)
0	34	55	120	1308, 331
1	50	51.5	131	1152, 320
2	66	48.5	150	1005, 305
2.7	78	46.5	150	950, 279
3	83	45.5	150	936, 291
4	101	44	150	899, 269

**Table 3 materials-14-02781-t003:** Trials to determine the optimum crumb rubber content for the 25% RAPB + 2.7% WFO.

CR. (%)	Penetration (PEN)	Softening Point (°C)	Ductility (cm)	Viscosity at 135 °C and 165 °C (cp)
1	74	47	129	961, 290
1.5	72	48	109	995, 300
2	67	50.5	101	1040, 329
2.5	61	52	88	1109, 375

**Table 4 materials-14-02781-t004:** Trials to determine the optimum waste frying oil content for the 40% RAPB.

WFO. (%)	Penetration (PEN)	Softening Point (°C)	Ductility (cm)	Viscosity at 135 °C and 165 °C (cp)
0	29	57	90	1702, 420
2	56	52.5	108	1422, 365
4	70	48.5	124	1106, 309
4.6	79	46	150	969, 281
6	92	44	150	953, 260
8	127	41.5	150	781, 229

**Table 5 materials-14-02781-t005:** Trials to determine the optimum crumb rubber content for the 40% RAPB + 4.6% WFO.

CR. (%)	Penetration (PEN)	Softening Point (°C)	Ductility (cm)	Viscosity at 135 and 165 °C (cp)
1.0	74	46.5	137	998, 293
1.5	70	48.0	102	1033, 302
2.0	61	51.5	90	1105, 337
2.5	55	52.5	65	1187, 411

**Table 6 materials-14-02781-t006:** Results of the virgin aggregate properties.

Aggregate Properties	Result	Criteria	Standard
Fine aggregate angularity (%)	53.5	Min 45	AASHTO T304 [38]
Deleterious materials (%)	0.27	0.2–10	ASTM C142 [39]
Flat and elongated particles in coarse aggregate (%)	3.89	Max 10	ASTM D4791 [40]
Los Angeles abrasion (%)	20	Max 35–45	AASHTO T 96 [41]
Aggregate impact value (%)	11.23	Max 30	BS812-112 [42]
Sand equivalent (%)	56	Min 45	ASTM D 2419 [43]
The specific gravity of aggregate	2.68	-	AASHTO T85 [44]
Specific gravity cement	3.15	-	AASHTO T133 [45]

**Table 7 materials-14-02781-t007:** Gradation for the 1200 and 2200 g asphalt mixture samples.

Sieve Size (mm)	Size 0.45	Percentage Passing (%)	Percentage Retained (%)	Weight of Sample (gm)	Weight of Sample (gm)
19	3.9	100	0	0	0
12.5	3.12	93.1	6.9	82.8	151.8
9.5	2.75	79.9	13.2	158.4	290.4
4.75	2.02	57.9	22	264	484
2.36	1.47	41.9	16	192	352
1.18	1.08	24.9	17	204	374
0.6	0.79	16	8.9	106.8	195.8
0.3	0.58	9	7	84	154
0.15	0.42	4	5	60	110
0.075	0.31	4	1	12	22
Pan	-	0	1	12	22
Cement	-	-	2	24	44
Total	-	-	100	1200	2200

**Table 8 materials-14-02781-t008:** Asphalt binder content of the RAP.

Sieve Size (mm)	Sample 1			Sample 2			Average Binder Content (%)	Calculated Binder Content (%)	Actual Binder Content (%)
	Before (g)	After (g)	Binder Content (%)	Before (g)	After (g)	Binder Content (%)			
Fine aggregate ≤2.36	500	473.2	5.36	500	470.5	5.9	5.63	4.88	4.9
Coarse aggregate ≥4.75	500	478.4	4.32	500	480.3	3.94	4.13		

**Table 9 materials-14-02781-t009:** Volumetric properties for all samples at the N design (100 gyrations).

Sample	Control	25% RAP	25% RAP + WFO + CR	40% RAP	40% RAP + WFO + CR	Superpave Criteria
Binder Content (%)	5.4	5.4	5.4	5.4	5.3	5.5
Va (%)	4.0	4.0	4.0	4.0	4.0	4.0
VMA (%)	15.9	14.9	15.1	14.6	15.5	Min 14%
VFA (%)	72.88	74.10	73.65	74.39	73.21	65–75

**Table 10 materials-14-02781-t010:** Moisture sensitivity test factors.

Parameter	Condition
Condition	Dry and saturated samples
Air voids	7 ± 0.5%
Saturation level	70–80%
Water bath period	24 h
Water path temperature	60 °C
T.S.R. requirement	≥80%

**Table 11 materials-14-02781-t011:** Parameters of the resilient modulus test.

Description	Value
Temperature	25 and 40 °C
Air voids	4 ± 0.5%
Preconditioning pulse	5
Load Cycle Time	3 s
Poisson’s	0.35 and 0.40
Force	1000 N
Rise Time	Vary
Pulse Repetition	1000 ms

**Table 12 materials-14-02781-t012:** Parameters for the dynamic creep test.

Parameter	Value
Temperature	40 °C
Air voids	4 ± 0.5%
Specimen height	Various
Specimen diameter	100 mm
Test Loading Stress	100 kPa
Pulse Width	100 kPa
Rest Period	900 ms
Contact Strain	10 kPa
No of Cycles	3600 cycles
Loading wave	Haversine

**Table 13 materials-14-02781-t013:** Parameters of the rutting depth test.

Parameter	Condition
Temperature (°C)	45 and 60 °C
Loading rate (cycles/min)	21 passes
Applied load (N)	520 N
Standard	BS 598-Part 110: 1998
Termination	After 45 min or depth over 15 mm

**Table 14 materials-14-02781-t014:** The significant difference in the group means for the ITS for the virgin asphalt with RAP and the recycled asphalt mixture.

Comparison of the Group Mean		Mean	Std. Error Mean	95% Confidence Interval of the Difference		Sig. (2-Tailed)
				Lower	Upper	
A—B1	Virgin 60/70—25% RAP	288.333	7.5351	−320.754	255.912	0.001
A—C1	Virgin 60/70—25% RAP + WFO + CR	33.66667	12.83658	−21.5647	88.89801	0.120
A—B2	Virgin 60/70—40% RAP	554.333	45.29287	−749.213	359.454	0.007
A—C2	Virgin 60/70—40% RAP + WFO + CR	60.66667	22.69606	−36.9866	158.3199	0.116

**Table 15 materials-14-02781-t015:** The significant difference in the group means between the moisture sensitivity of the virgin asphalt with RAP and the recycled asphalt mixture.

Comparison of Group Mean		Mean	Std. Deviation	95% Confidence Interval of the Difference		Sig. (2-Tailed)
				Lower	Upper	
A—B1	Virgin 60/70—25% RAP	3.61667	4.13512	−6.65555	13.88888	0.269
A—C1	Virgin 60/70—25% RAP + WFO + CR	0.68667	0.9609	−1.70035	3.07368	0.341
A—B2	Virgin 60/70—40% RAP	10.88333	3.80865	1.42212	20.34455	0.038
A—C2	Virgin 60/70—40% RAP + WFO + CR	2.19333	4.63768	−9.32729	13.71396	0.499

**Table 16 materials-14-02781-t016:** The significant difference in the group means between the virgin and RAP mixture and the recycled asphalt mixture for the resilient modulus at 25 °C.

Comparison of the Group Mean		Mean	Std. Deviation	95% Confidence Interval of the Difference		Sig. (2-Tailed)
				Lower	Upper	
A—B1	Virgin 60/70—25% RAP	3727.17	246.1198	–3985.45	3468.88	0.000
A—C1	Virgin 60/70—25% RAP + WFO + CR	92.16667	222.6077	–141.446	325.7791	0.357
A—B2	Virgin 60/70—40% RAP	4921	202.0277	–5133.02	4708.98	0.000
A—C2	Virgin 60/70—40% RAP + WFO + CR	69	151.2706	–227.749	89.74878	0.315

**Table 17 materials-14-02781-t017:** The significant difference in the group means for the resilient modulus of the virgin asphalt with RAP and the recycled asphalt mixture at 40 °C.

Comparison of the Group Mean		Mean	Std. Deviation	95% Confidence Interval of the Difference		Sig. (2-Tailed)
				Lower	Upper	
A—B1	Virgin 60/70—25% RAP	812.833	107.4401	−925.585	700.082	0.000
A—C1	Virgin 60/70—25% RAP + WFO + CR	41.16667	93.96471	−57.4433	139.7766	0.332
A—B2	Virgin 60/70—40% RAP	1524	125.5388	−1655.74	1392.26	0.000
A—C2	Virgin 60/70—40% RAP + WFO + CR	11.8333	53.10901	−67.5678	43.90115	0.609

**Table 18 materials-14-02781-t018:** The significant difference in the group means for the dynamic creep of the virgin asphalt with RAP and the recycled asphalt mixtures at 40 °C.

Comparison of the Means		Mean	Std. Deviation	95% Confidence Interval of the Difference		Sig. (2-Tailed)
				Lower	Upper	
A—B1	Virgin 60/70—25% RAP	0.32667	0.02333	0.22627	0.42706	0.005
A—C1	Virgin 60/70—25% RAP + WFO + CR	0.02333	0.01764	−0.05256	0.09922	0.317
A—B2	Virgin 60/70—40% RAP	0.41167	0.02682	0.29626	0.52707	0.004
A—C2	Virgin 60/70—40% RAP + WFO + CR	0.04333	0.02906	−0.0817	0.16837	0.274

**Table 19 materials-14-02781-t019:** The significant difference in the group means for the rutting of the virgin asphalt with RAP and the recycled asphalt mixture at 45 °C.

Comparison of the Group Mean		Mean	Std. Deviation	95% Confidence Interval of the Difference		Sig. (2-Tailed)
				Lower	Upper	
A—B1	Virgin 60/70—25% RAP	0.73667	0.25106	0.11299	1.36035	0.037
A—C1	Virgin 60/70—25% RAP + WFO + CR	0.04	0.26514	−0.69865	0.61865	0.818
A—B2	Virgin 60/70—40% RAP	1.29667	0.09815	1.05285	1.54048	0.002
A—C2	Virgin 60/70—40% RAP + WFO + CR	0.00667	0.17616	−0.44428	0.43095	0.954

**Table 20 materials-14-02781-t020:** The significant difference in the group means for the rutting of the virgin asphalt with RAP and the rejuvenated asphalt mixture at 60 °C.

Compare Group Mean		Mean	Std. Deviation	95% Confidence Interval of the Difference		Sig. (2-Tailed)
				Lower	Upper	
A—B1	Virgin 60/70—25% RAP	0.89	0.20075	0.39131	1.38869	0.017
A—C1	Virgin 60/70—25% RAP + WFO + CR	0.03667	0.14978	−0.3354	0.40873	0.713
A—B2	Virgin 60/70—40% RAP	1.60333	0.21502	1.0692	2.13747	0.006
A—C2	Virgin 60/70—40% RAP + WFO + CR	0.07	0.12767	−0.24715	0.38715	0.443

## Data Availability

All data used in this research can be provided upon request.

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
