# Peer review of "Mechanical Performance of Reclaimed Asphalt Pavement Modified with Waste Frying Oil and Crumb Rubber"

_materials, 2021, doi:10.3390/ma14112781_

Round 1
Reviewer 1 Report
The topic of this paper is well related to the theme of the special issue of the journal. In general, this paper presents a valid laboratory research on use of waste materials (i.e., waste frying oil and crumb-rubber) as rejuvenating agents in hot asphalt mixtures containing reclaimed asphalt pavement. The use of waste frying oil and crumb-rubber as rejuvenating agent is not novel concept but presented research may have significance in promoting use of waste and recycled materials in local community.
I suggest accepting the paper after a minor revision.
Specific comments:
- Please check the language, as there are a few places where there were grammatical and editorial errors (for example the use of a phrase ' In the meantime').
- For better flow of the paper, it is suggested to move the section ‘3.1. Optimum asphalt binder content’ to section 2, as sub-section 2.1.5. This is suggested because, optimum asphalt binder content is input parameter in the design of asphalt mixtures and as such it belongs to the section relating to the materials. Section ‘3. Results and discussion’ should contain only the performance test results and the corresponding discussion with sub-sections 3.1. Indirect tensile strength; 3.2. Moisture sensitivity; etc.
- Section 2.1. it is stated ‘The WFO and CR percentages were selected based on the trials using physical tests such as penetration, softening point, ductility, and viscosity.’ were these percentages adopted from some previous study? If so, it should be referenced. If not, it is suggested to add this research as additional sub-section.
- Sub-section 2.1.3. what was the heating temperature? The heating temperature should be specified.
- In section ‘3.1. Optimum asphalt binder content’ the OBC is performed for all five asphalt mixtures but in table 9. results are shown for three mixtures (Virgin 60/70, 25% RAP + WFO + CR, 40% RAP + WFO + CR). Add results of OBC for 25% and 40% RAP mixtures.
- For better clarity it is suggested to add table with asphalt mixtures label (e.g. Virgin 60/70, 25%RAP, 25% RAP + WFO + CR etc.), optimum bitumen content, RAP and WFO content
- Were RAP, WFO and CR added as mass or volume percentage?
- Sub-section 2.2.2. please consider making it shorter since the test is done by well known standard it is not necessary to describe it in such detailed manner.
- Please explain the reasons for using t-test. Add a description of labels A, B1, B2, C1 and C2.
- Please make sure that conclusions section underscores the scientific value of the paper focusing on the difference between the RAP mixtures with and without rejuvenating agent, and similarities between virgin mixture and RAP mixture with rejuvenating agent for all performance tests.
Author Response
Thank you for your positive, fruitful comments and suggestions, which have improved our manuscript's quality. Please find below is the revision report for your attention and perusal. The responses have been arranged based on your feedback in the review process.

Reviewer 2 Report
The paper describes an experimental investigation concerning the comparison between a reference hot mix asphalt and HMAs that include waste frying oil and crumb rubber.
The topic is interesting and the tests performed can be considered appropriate. However, the experimental program is not properly described and some comments to the results should be correct.
An English revision is also necessary.
Some suggestions can be found below:
- At page 3, line 105, you wrote that “The RAP was heated at 105 °C in an oven for 24 h to remove moisture content” but at page 5, line 171 you wrote: “1hr was selected as the heating period for all the RAP amounts used in this study”. Please, explain better why you used different temperatures.
- Page 3, table 1 – From this table it seems that you studied two percentages of WFO content (2.7 and 4.6%) but in the results it seems that only one percentage was tested. Please, specify better the experimental program.
- Figure 1 – Which temperature was used for obtaining this figure?
- Table 4 – I would like to suggest using, in the future, the two different gradations of the RAP separately as the granulometric curve of the HMA will result more precise.
- 2.2 – The experimental program is not correctly reported. Until this point of the paper, you described the materials but not the composition of the mixtures that you compared (i.e. WFO content and RAP content). You just wrote something about the percentages of RAP and WFO but not the combination. Considering Table 1, I can think that you are going to compare all the possible combinations of the components: 25%RAP, 25%RAP+2.7%WFO, 25%RAP+2.7%WFO+ 1.5%CR, 25%RAP+4.6%WFO, 25%RAP+4.6%WFO+ 1.5%CR, 40%RAP, … and so on. On the contrary, from Figure 2, I understood that you considered only one percentage of WFO. Which percentage?
Please, correctly write the experimental program by including also the number of repetitions for each test (only for the “moisture susceptibility” test you specified “two sets of three”)
- Page 6, line 215 – You wrote “The samples were cured in an oven at 60°C for 16 hours and 135°C for 2 hours before compacted the field's required level using the SGC”. Is “the field’s required level” referred to the 7% air voids shown in Table 5? Please, specify.
- What about the air void content of the specimens tested with the other test methods (resilient modulus, dynamic creep test and wheel tracking test)?
- Page 8, line 270 – You wrote: “dimension of 300 × 300 × 50 mm and height of 65±1 mm”. What is the shape of the specimens? The fourth dimension is incorrect.
- Table 6 and Table 7 – How many specimens did you test for each temperature? And what about the air void content of these specimens?
- Page 8, line 288 – You write: “five asphalt mixtures (Virgin 60/70, 25% RAP + WFO + CR, 40% RAP + WFO + CR)”. You write “five” but in the brackets you included three mixtures. If you specify better in the experimental program, as suggested above, this became cleared.
- Page 9, lines 294-304. In this part of the text you have repeated the same concept too many times. You used 11 lines to say a vey simple concept, please summarize.
- Figures 2, 3, 4, 5 and 6 – Please, add error bars.
- Tables 10, 11, 12, 13, 14, 15 and 16 – What does A, B1, B2, C1 and C2 mean?
Moreover, the comments related to these tables can be shortened.
- Page 11, lines 381-383 – Please check this sentence
- 3.2.3 – You performed resilient modulus tests but in Figure 4 showed the stiffness modulus results and commented everything in terms of stiffness modulus. Since resilient modulus tests were performed, Figure 4 should show resilient modulus results. Also, the comments should be in terms of resilient modulus. Moreover, the cited papers (e.g. reference [15]) refer to stiffness modulus tests and not to resilient modulus test. Please, be consistent with the tests performed.
Page 11, line 398. “8930” should be “8390”.
- 3.2.3, §3.2.4 and § 3.2.5 – All comments on the results should be shortened. It is not necessary to repeat in the text all the numbers that are already in the figures.
On the contrary, it would be interesting to compare the results of Figure 5 and Figure 6 since they are both related to the permanent deformation properties of the materials.
- Page 16, lines 545-548 – You wrote: “Figures 6 and 3 shows the relation was the inverse relationship between the wheel tracking and moisture damage results using WFO as a rejuvenator. This result was consistent with a previous study performed by Yang et al. [43], in which the increase in the rutting resistance reduced the TSR”. These two parameters are not strictly linked as you wrote (“the increase in the rutting reduced the TSR”). It would be better to highlight that a higher moisture sensitivity should provide a lower rutting resistance, as your study also confirmed.
- In the conclusions, you used very often the term “asphalt combination” but it is not clear what it means as it has never been used in the text. The conclusions are not correctly written and should be re-written by underlying the main results obtained.
- Page 17, line 578. The rutting resistance for the recycled asphalt incorporating waste materials is almost equal to that for the conventional asphalt, diversely to what you wrote. Error bars will likely show this.
Author Response

(The authors gave the same response as above.)

Reviewer 3 Report
What is new in this research? I can`t find anything new.
Author Response

(The authors gave the same response as above.)

Round 2
Reviewer 2 Report
The Authors accepted my revisions, tried to change the paper according to them but did not dissolve all my doubts.
Comment 1 – It is still unclear why two different procedures were followed to heat the RAP
Comment 2 – Authors added Tables 2 and 4 but did not comment on the criteria for choosing the percentages of WFO and CR and thus it is still impossible to understand their choice.
Comment 5 – The criteria for choosing the percentages of WFO and CR are still unclear
Comment 6 – Authors did not answer to my question: is “the field’s required level” referred to the 7% air voids shown in Table 5?
Comment 9 – Please, add in the text the information that you wrote in the answer (coverletter)
Comment 12 – Please, check the error bars. They are identical for all the materials and this seems “strange”. For example, the error bars in Figure 3 are identical for all the five materials, both for dry and wet conditions, and this is almost impossible. And this is true for all the figures.
Comment 15 – The sentence “According to Poulikakos et al. [15], this trend indicates that higher test temperatures resulted in lower stiffness of the asphalt mixture” should be replaced with “According to Poulikakos et al. [15], higher test temperatures resulted in lower stiffness of the asphalt mixture”.
Comment 19 – Please, specify what you mean for “recycled asphalt containing waste materials”. I understood 25%RAP+WFO+CR and 40%RAP+WFO+CR. If this is the case, you can observe from Figure 6 that the rut depth for Virgin 60/70, 25%RAP+WFO+CR and 40%RAP+WFO+CR are 4.53, 4.49 and 4.45 mm, respectively. They are very similar and you cannot say that “The rutting resistance of the recycled asphalt containing waste materials was higher than the virgin asphalt at 60 °C”. From Table 19 you can also see that they are not significatively different at 45 °C and if you will repeat the calculation for 60 °C you will probably obtain the same non-significant difference. Moreover, check the error bars since they are probably incorrect.
Author Response
Review Report Comments for Reviewer 2
Manuscript ID: Materials-1183300
Thank you for your positive, fruitful comments and suggestions, which have improved our manuscript's quality. Please find below is the revision report for your attention and perusal. The responses have been arranged based on your feedback in the review process.
|
Comments |
Amendments |
Location of the additional write up |
|
Comment 1 It is still unclear why two different procedures were followed to heat the RAP |
Thanks for the kind observation. It is only one procedure was done to heat the RAP but it was mentioned twice in sections 2.1. and 2.1.4. The sentence in section 2.1. was deleted and it kept the whole process of heating the RAP in section 2.1.4. |
Section 2.1.4 |
|
Comment 2 Authors added Tables 2 and 4 but did not comment on the criteria for choosing the percentages of WFO and CR and thus it is still impossible to understand their choice. |
Your comments are appreciated. For the WFO, the asphalt binder Needs a high percentage of WFO to pass all the requirements of the physical tests. For the CR, the asphalt binder needs a proper percentage of CR to increase the stiffness with suitable workability. |
|
|
Comment 5 The criteria for choosing the percentages of WFO and CR are still unclear |
Thanks again for the valuable comment. The criteria for choosing the percentages of WFO and CR were added in section 2.1.1 as follows: “The percentages of the WFO were chosen depending on the criteria of the physical tests with a lower viscosity result. On the other hand, the percentages of the CR were chosen depending on the stiffness with appropriate workability”. |
Lines 104-107 |
|
Comment 6 Authors did not answer to my question: is “the field’s required level” referred to the 7% air voids shown in Table 5? |
Thanks for the question. No, it is not referred. And the sentence was rewritten as follows: “The samples were cured in the oven at 60°C for 16 hours and 135°C for two hours before compacted by using the Superpave Gyratory Compactor (SGC)”. |
Line 212-214 |
|
Comment 9 Please, add in the text the information that you wrote in the answer (coverletter) |
Your comments are appreciated. The sentences were improved as follow: “A total of 105 specimens for tests in terms 3 specimens for ITS test, 6 specimens for moisture sensitivity tests ( 3 for conditions specimens (Wet) and 3 for un-condition (Dry) specimens, 3 specimens for dynamic creep, 6 specimens for resilient modulus test (3 specimens for 25°C and 3 specimens for 40°C). A 110 specimens for the process of OBC, air voids and test conditions”. |
Section 2.2 Lines 186-191 |
|
Comment 12 Please, check the error bars. They are identical for all the materials and this seems “strange”. For example, the error bars in Figure 3 are identical for all the five materials, both for dry and wet conditions, and this is almost impossible. And this is true for all the figures. |
Thanks for the kind observation. The Figures were edited. |
Figures 2, 3, 4, 5, and 6 |
|
Comment 15 The sentence “According to Poulikakos et al. [15], this trend indicates that higher test temperatures resulted in lower stiffness of the asphalt mixture” should be replaced with “According to Poulikakos et al. [15], higher test temperatures resulted in lower stiffness of the asphalt mixture”. |
Thanks for the kind observation and suggestions for improvement. The sentence was edited to be as follows “According to Poulikakos et al. [15], higher test temperatures resulted in lower stiffness of the asphalt mixture”. |
Lines 341-342 |
|
Comment 19 Please, specify what you mean for “recycled asphalt containing waste materials”. I understood 25%RAP+WFO+CR and 40%RAP+WFO+CR. If this is the case, you can observe from Figure 6 that the rut depth for Virgin 60/70, 25%RAP+WFO+CR and 40%RAP+WFO+CR are 4.53, 4.49 and 4.45 mm, respectively. They are very similar and you cannot say that “Th e rutting resistance of the recycled asphalt containing waste materials was higher than the virgin asphalt at 60 °C”. From Table 19 you can also see that they are not significatively different at 45 °C and if you will repeat the calculation for 60 °C you will probably obtain the same non-significant difference. Moreover, check the error bars since they are probably incorrect. |
Your invaluable comments are appreciated. The mistake was Corrected as follows: “The rutting resistance of the recycled asphalt containing waste materials was slightly improved compared to the virgin asphalt at 60 °C”. |
Lines 506-508 |
Thank you for your kind attention and cooperation. We appreciate all your comments and feedback.

Reviewer 3 Report
Thank you for responding to my comment. As I worried, you do not have new findings as for scientific publication. It is a nice technical report, where you repeated other people work and used specific materials to determine specific 1.5% of CR will improve the overall performance. It is not a general conclusion I would expect in a scientific publication. What will happen if I change materials, will it still be 1.5%?
Author Response

(The authors gave the same response as above.)
